# Object Play as a Positive Emotional State Indicator for Farmed Spotted Paca (*Cuniculus paca*)

**DOI:** 10.3390/ani14010078

**Published:** 2023-12-25

**Authors:** Allison F. de Lima, Stella G. C. Lima, Sérgio L. G. Nogueira-Filho, Suzanne D. E. Held, Michael Mendl, Selene S. C. Nogueira

**Affiliations:** 1Applied Ethology Laboratory, State University of Santa Cruz, Ilhéus 45662-900, BA, Brazil; allisonlimazootecnista@gmail.com (A.F.d.L.); stellagcl@hotmail.com (S.G.C.L.); slgnogue@uesc.br (S.L.G.N.-F.); 2Animal Welfare and Behaviour Group, Bristol Veterinary School, University of Bristol, Bristol BS8 1QU, UK; suzanne.held@bristol.ac.uk (S.D.E.H.); mike.mendl@bristol.ac.uk (M.M.)

**Keywords:** animal welfare, applied ethology, animal emotions, farmed animals, play behavior, positive emotional state indicator

## Abstract

**Simple Summary:**

The spotted paca (*Cuniculus paca*) has been legally bred by small-scale producers in Brazil as an alternative source of protein. Although captive breeding of this species is considered relatively easy and promising for farmers, little is known about its welfare in captivity. Therefore, using boomer balls to induce object play, we investigated whether object play behavior can be used as a positive emotional state indicator for spotted pacas by examining whether it correlated with other pre-validated positive welfare markers, such as affiliative behavior and low amplitude bark vocalizations. As expected, we found that boomer balls stimulated play. At the same time, the spotted pacas showed more affiliative and exploratory behaviors, with decreased occurrence of agonistic interactions. We also found an increase in barking with low mean amplitude when the paca were provided with boomer balls. Object play behavior thus seems to be a promising non-invasive indicator of positive emotional state in this species because it was associated with an increase in low amplitude barks and more affiliative behavior. As object play can also improve welfare, stimulating its expression, through the provision of boomer balls, should be encouraged on spotted paca farms.

**Abstract:**

We aimed to assess whether object play can be used as a positive emotional state indicator for farmed spotted pacas (*Cuniculus paca*) by examining its association with other positive welfare markers including affiliative behavior and low-amplitude vocalizations. We submitted six groups of spotted pacas (one male/two females per group) (N = 18) to an ABA experimental design (A_1_/A_2_: without ball; B: with three boomer balls). Object play behavior occurred only during phase B (mean = 35.5 s, SE = 6.4). The spotted pacas spent more time in affiliative and exploratory behaviors and less time engaging in agonistic interactions during phase B than in both control phases (A_1_ and A_2_) (*p* < 0.05). Moreover, the spotted pacas emitted more low-amplitude bark vocalizations during phase B than during either control phase (*p* < 0.05), and such vocalizations have previously been shown to indicate a positive affective state and low arousal level. Because the expression of object play was associated with a decrease in aggression, an increase in affiliative behavior, and an increase in low-amplitude barking, we suggest that object play can be used as a non-invasive indicator of positive emotional state in this species.

## 1. Introduction

‘Play’ covers a range of behaviors with shared characteristics such as an apparent lack of an immediate goal and specific body movements (play markers, e.g., [1]) and is often expressed in the absence of fitness threats (e.g., [2]). Play behaviors can be expressed solitarily or in a group [3]. Three types are distinguished: locomotor play (running, hopping, leaping, etc.), social play (playing with another), and object play (playing with objects). Burghardt [2] states five characteristics that distinguish a behavior as being ‘play’: “*(a) this behavior is incompletely functional in the context expressed, (b) play is voluntary or rewarding, (c) it is modified developmentally or structurally compared with when it is used in its normal, functional context, (d) performed repeatedly, but not necessarily in an invariant form and (e) that it starts in healthy, relatively unstressed animals in a relaxed context*”.

Some of these characteristics have led researchers to suggest play as a promising welfare indicator, while recognizing that play can generate, as well as reflect, improved or good welfare [4,5,6,7]. Play behavior is easy to detect and measure in a practical and non-invasive way [2,8]. Another peculiar characteristic of this behavior is its tendency to be contagious as when initiated by one or two individuals, it tends to spread to other individuals in the group [9,10]. Therefore, stimulating the occurrence of play behavior in captive animals has the potential to improve their current welfare and spread improved welfare in a group [5], as has been observed in some species using environmental enrichment (e.g., *Tayassu pecari* [11]; domestic pigs [12]). To validate the use of play as a positive emotional state indicator, it is first necessary to verify its association with other pre-validated markers of improved welfare, such as the increased expression of affiliative and exploratory behaviors or some specific calls [4].

Affiliative interactions, characterized by close spatial proximity and the exchange of social-positive behaviors, promote group cohesion through the formation of bonds between animals [13,14]. They can thus reflect positive welfare [15]. Vocalizations can also be used as markers of positive welfare. For example, in rats (*Rattus norvegicus*), 50 kHz calls are associated with positive appetitive behaviors such as play and mating, while 22 kHz calls are associated with negative affective behaviors such as biting at the start of a fight [16]. Establishing links between play behavior and these other types of welfare indicator can help validate play as a positive emotional state marker in species about which little is known and whose domestication process is still in its infancy, such as the spotted paca (*Cuniculus paca*).

The spotted paca is the second largest rodent in the world [17], after the capybara (*Hydrochoerus hydrochaeris*). Considered an important seed disperser, this species is frugivorous, feeding on fruit flesh and seeds that fall to the forest floor [18]. The spotted paca is thought to be nocturnal, solitary, territorial, burrowing, and aggressive toward conspecifics, with a monogamous or promiscuous mating system [19,20,21]. However, the presumed solitary habits of this species are questionable. Although an adult spotted paca usually forages alone or with its young at night, small groups may forage together in areas of abundant food [22]. Moreover, according to territory use, the social system in the species appears to be more flexible than originally expected [23]. Spotted paca may have overlapping home ranges, with neighboring individuals sharing food. This depends on the availability of resources such as food, suitable sites for burrows and water pools, and predation/hunting pressure [23]. Furthermore, Lima et al. [24] described a vocal complexity in spotted paca communication similar to that of group-living species. This finding may explain the relative ease with which farmers can raise spotted pacas in groups [19].

Spotted paca are one of the most hunted neotropical animals due to widespread appreciation of their meat [19,25,26]. In Brazil, the spotted paca is farmed legally, mainly by small producers [27]. Although captive breeding of the species is considered relatively simple and promising for farmers, there is very little known about the welfare of this neotropical species in captivity (e.g., [28,29,30,31]). In a recent study, Lima et al. [30] found that some vocalizations in adult pacas of both sexes are linked to their affective state and level of arousal. Specifically, the authors found that in a positively valenced context (feeding time), pacas emitted many more bark calls with lower mean amplitudes and a shift in the third quartile frequency (Q75) to a lower frequency compared with those emitted in a negative context (pen cleaning) [30]. Lima et al. [30] also verified that the mean amplitude of the bark was higher when spotted pacas were at high arousal levels compared with low arousal levels. Based on these results, these authors suggested that the decreases in both the mean amplitude and energy distribution (Q75) of bark calls are associated with positive emotional valence for the spotted paca, while an increase in the mean amplitude of the bark call is associated with negative conditions and an increase in arousal levels in this species.

A lack of attention to the needs of captive-bred animals can lead to chronic stress and abnormal behaviors, such as stereotypies that compromise their welfare [1,32] and productivity [33,34]. On the other hand, play behavior may reflect positive welfare, although this relationship is not always straightforward (e.g., [5,6]). In spotted paca, play has so far been reported only in captive young animals up to two months old [28], and locomotor play, in which the animals run alone in the enclosure and sometimes shake their heads, was reported. The aim of this study, therefore, was to evaluate whether object play also occurs in adult paca, and whether it might be used as a positive emotional state indicator for farmed spotted paca. This was evaluated by investigating its association with other behaviors that have been previously found to be associated with positive welfare in pacas.

If object play indicates positive affective states and welfare as proposed by Lawrence [35], we would predict an increase in pre-validated ‘positive’ behaviors, such as affiliative and exploratory behaviors, as verified in other species (e.g., *Tayassu pecari* [11]), and reduced aggression, as observed in domestic pigs following environmental enrichment [36]. If an increase in bark calls with lower mean amplitude and a shift in the energy distribution (Q75) toward lower frequency indicates a positive affective state in spotted pacas [30], we expect more bark calls with lower mean amplitude and lower frequency of Q75 when the spotted pacas are playing with objects than when they are not.

## 2. Materials and Methods

### 2.1. Ethical Note

This work was conducted in accordance with Brazilian laws and was approved by the Ethics Committee on Animal Use (CEUA) of the State University of Santa Cruz (protocol # 029/18).

### 2.2. Subjects and Housing Conditions

The experiment was conducted at a farm located near the city of Soledade de Minas, in the state of Minas Gerais, Brazil. We recorded data from 18 adult spotted pacas (*Cuniculus paca*) (12 females and six males), born and raised in captivity, aged from one to four years, and kept in six groups (one male and two females per group). The spotted pacas were identified by natural characteristics, such as scars and coat color, without requiring additional marking. The animals were housed in 4 m^2^ pens with cement floors covered with ceramic tiles. The pens were surrounded by 0.5 m high brick walls with a 2.0 m high wire mesh above the walls. In each pen there was a wooden shelter (1.5 m long × 1.5 m wide × 1.0 m high), a water tank (0.6 m long × 0.3 m wide × 0.3 m high) that provided water *ad libitum* and three feeders (0.4 m long × 0.3 m wide × 0.3 m high; one feeder per animal). The keeper provided food at the three feeders at around 16:00 h each day. The meal consisted of corn meal (150 g per animal), seasonal fruits and vegetables. The keeper cleaned the pens daily (7:00 h), sweeping the floor and shelter, and washing the feeders and water tank.

### 2.3. Data Collection

A single observer (SGCL) recorded the behavior of spotted pacas (Table 1) using a camcorder (Sony HDR-CX240, Manaus, Brazil), which was fixed on a tripod. We set up the camcorder outside the enclosure at about 2.0 m from the animals. The spotted pacas were habituated to the presence of the observer for a period of seven consecutive days prior to data collection. After habituation, the observation of the groups and the order of the animals was drawn daily by lot. The behavior duration of each individual in the groups was observed for 10 min per day over three consecutive days for all phases of the study (described below). Thus, a total of 27 h of data collection was conducted for all six groups, with 9 h per phase. Further details are provided below. The observation sessions took place between 16:00 h and 18:00 h (before feeding), using animal focal sampling [37].

We used the ABA paradigm [43]: phases A_1_ and A_2_ corresponded to control phases with an absence of objects used to motivate spotted pacas to play (boomer balls), while during phase B, we introduced objects (boomer balls) to motivate animals to play. The introduction of boomer balls was necessary, because adult pacas do not usually show play behavior spontaneously. Thus, in phase B we introduced three boomer balls to each group for 30 min daily. The boomer balls were made of hard plastic material and had a diameter of 0.15 m. Aiming to avoid competition and encourage play behavior, each individual was given one ball. The animals used in this study had no prior experience with balls. The choice of boomer balls was made because in a previous study, Nogueira et al. [29] found that spotted paca interacted with them during a novel object temperament test, chasing the ball in a way that met all of Burghardt’s [2] criteria for play behavior. Furthermore, balls have not been shown to be a threat to animals and are one of the most highly valued objects for inducing play behavior in mammals [3]. After collecting the data, another observer (AFL) analyzed the recorded footage using CowLog software version 3.0.2 [44].

### 2.4. Acoustic Parameters of Bark Calls

The same observer (SGCL) recorded animal calls (Table 1) using a Sennheiser ME-66 unidirectional microphone (Wedemark, Germany) and a Tascam digital recorder (model: DR-100 MK II, settings: WAV format, mono mode, 48 kHz sampling rate and 16-bit resolution). The observer recorded the bark calls emitted by the animals without interruption, keeping the recorder on until the end of each call, in accordance with the method of Lima et al. [24,30]. Briefly, another observer (AFL) was given a list of bark calls to analyze along with the corresponding timestamps in the video footage to conduct this analysis. The observer could easily identify the caller because of their natural characteristics, as explained above. For acoustic analysis, another researcher (SLGNF), blinded to experimental phases and animal identity, selected only the high-quality bark calls without background noise and/or overlap. Therefore, from a total of 261 recorded bark calls, the observer selected 257 bark calls of better quality for the analysis of acoustic parameters. Nevertheless, to compare the occurrence of call types between the experimental phases, all emitted vocalizations were considered, regardless of their acoustic quality.

We used Raven Pro version 1.5 software (Cornell Lab of Ornithology, Ithaca, NY, USA) to measure the third frequency quartile (Hz) (Q75: frequency value at the upper threshold of the third energy quartile) [45] of barks emitted by the pacas in all experimental phases. Additionally, we used Praat software version 5.3.06 [46] to measure the mean amplitude (dB) [47] of these barks. For this analysis, we applied the following settings: time range: FFT method; window size: 0.01; time range: 1000; frequency range: 250; Hanning window shape; dynamic range: 60 dB.

### 2.5. Data Analysis and Statistics

We compared the time animals spent in each of the recorded behaviors (see Table 1) across experimental phases using mixed linear models (MLMs), applying one model for each behavior (affiliative, agonistic, and exploratory). The models were fitted to the data using the restricted maximum likelihood (REML) method. Because play behavior only occurred during the ball phase (B), play could not be compared between phases. In MLMs, we considered as fixed factors the sex (male and female) and the experimental phases (controls: A1 and A2; ball phase: B), as well as their interaction. When the interaction was significant, we used Tukey’s post hoc tests. We also used MLMs to compare the acoustic parameters (mean amplitude and Q75) of the barks emitted by the pacas. In the models used to analyze the acoustic parameters, only the experimental phases were included as a fixed factor. In all models, the identity of each individual nested within its group was included in the models as a random factor. This allowed us to control for repeated measures dependencies. We graphically checked the residuals of each model for a normal distribution and homoscedasticity, and used logarithmic transformations for all but the mean amplitude parameter to satisfy these assumptions. We used the chi-square goodness-of-fit test to compare the barking of the pacas between experimental phases. We used Minitab 21.1 software (Minitab Inc., State College, PA, USA) for all analyses, considering *p* < 0.05 as significant.

## 3. Results

Object play only occurred during the environmental ball phase (B), with a mean duration of 35.5 s (standard error (SE) = 6.4), regardless of the subtypes of object play (Figure 1d). The pacas displayed six different subtypes of object play (Table 1). Biting was the most frequent subtype of object play (Table 2). There was no difference in the amount of time females (mean = 13.3-s, SE = 1.4) and males (mean = 24.0-s, SE = 1.4) played with the balls (Test-t = 1.29, *p* = 0.21).

We recorded affiliative behaviors only in the first control phase (A_1_) and in the ball phase (B). No affiliative behavior occurred in phase A_2_. Pacas were observed to spend longer performing affiliative behaviors in phase B (Table 3 and Figure 1a). There was also variation in the expression of exploratory behavior and agonistic interactions depending on the experimental phase (Table 3). The post hoc tests showed that pacas were observed to spend longer performing exploratory behavior during the ball phase (B) than in the control phases (A_1_ and A_2_) (Figure 1b). On the other hand, pacas interacted agonistically for less time during the ball phase (B) than during the control phases (A_1_ and A_2_) (Figure 1c). Among the subtypes of object play, they spent more time biting than carrying or touching and biting the boomer balls (Figure 1d). Sex and the interaction between sex and experimental phase did not affect the time the pacas were observed in the behaviors analyzed (Table 3). However, during the ball phase (B), males showed a trend (F_1, 7.83_ = 4.94, *p* = 0.058, Table 3) to display affiliative behavior for a longer duration than females (males: mean = 20.8-s, SE = 9.0; females: mean = 6.9-s, SE = 2.0).

We found an increase in bark emissions during the ball phase (B) (Χ^2^ = 311.06, DF = 2, *p* < 0.001, Figure 2). We also found an effect of the experimental phase on the mean amplitude of the barks (F_2, 204.64_ = 9.03, *p* < 0.001). The post hoc tests showed that during the ball phase (B), the pacas emitted barks with a lower mean amplitude than in the two control phases (A_1_ and A_2_) (Figure 3a). On the other hand, the frequency in the third quartile (Q75) of barks did not differ between experimental phases (F_2, 156.6_ = 2.24, *p* = 0.110) (Figure 3b).

## 4. Discussion

The presence of the boomer balls in the spotted pacas’ enclosures motivated play behavior, increased the time spent on affiliative and exploratory behaviors, and reduced the frequency of agonistic behaviors. The amount of barking with a lower mean amplitude was increased in the presence of play behavior, suggesting that playing with balls is a positive event, corroborating our hypothesis. However, the expected lowering of the energy distribution (Q75) of barks towards a lower frequency, also thought to indicate a positive state [30], was not observed.

The time the pacas spent interacting with the boomer ball showed that this object was an attractive stimulus for the animals, generating object play involving manipulation with the mouth and/or paws. We considered such interactions of paca with the boomer ball to be consistent with Burghardt’s [2] five characteristics that distinguish a behavior as being in a “play” state. The interaction with the boomer ball was completely non-functional in the context in which it was expressed. Such interaction was voluntary, performed repeatedly along with some variant forms, and started in healthy and relatively unstressed animals in a relaxed context. We observed that individuals sometimes carried the boomer balls into the burrow. This behavior may be related to competition for the object as a limited resource and thus movement away from others to avoid disputes. Solitary play behavior is common in several species, including those that are social, such as the *Tayassu pecari* [11]. For example, in this species, as expected, adding objects to the environment promotes increased possession of the objects by the dominant individual early in the animals’ exposure to the new objects, and later the objects are shared by the dominant individuals [11]. Solitary play facilitates the acquisition of new behavioral patterns, such as in long-tailed Burmese monkeys (*Macaca fascicularis aurea*), whose juveniles improve their foraging skills through solitary play with objects [48]. Furthermore, solitary play is important in a human-controlled environment because it allows individuals to exercise control over their own activity pattern [49].

We found no effect of sex on the expression of object play, affiliative, exploratory, or agonistic behaviors in spotted pacas, suggesting that males and females were equally interested in the boomer ball. Sex differences are well known for social play, specifically play fighting in rodents, but less investigated to date for object play. In rats and hamsters, some authors have reported that males display more social play than females, while others have reported no sex difference and others have suggested that these sex differences are related to experimental conditions (for a comprehensive review, see Cooper et al. [50]). In adult spotted pacas, males and females can be distinguished by the head size [51]. The skull of pacas has an expanded zygomatic arch in males, making their heads larger than those of females [52]. This head characteristic is reflected in the differences in some of their vocal parameters [24]; however, our data do not reveal any association with object play and other behavior evaluated herein.

As we predicted, affiliative behaviors were enhanced in the boomer ball phase, favoring the occurrence of positive social interaction in the group, which indicates a beneficial social condition for the pacas in our study. Increased affiliative interactions are indicative of increased group cohesion [13], due to the formation of bonds between animals, which are characterized by physical approaches, and can express positive welfare [15]. Thus, the increase in affiliative behaviors suggests that during the period when the boomer balls were present, the animals were in a more positive state. Nevertheless, it seems reasonable that the absence of balls in phase A_2_, after three consecutive days of playing with this type of enrichment, may have negatively affected the animals’ mood. This may explain the lack of affiliative interactions during this phase.

Exploratory behaviors also intensified in the enriched ball phase compared with the control phases. One of the important properties of an enrichment object, such as a boomer ball, is to generate novelty and promote an increase in general activities, including exploration [53]. Increasing activities for animals may contribute to reducing inactivity, which itself may be associated with a state of distress [54]. Exploratory behavior is desirable in captive animals [55], suggesting that promoting this behavior contributes to improved animal welfare [4]. We can consider, therefore, that in our study there was improvement in the pacas’ welfare because their play and exploratory behaviors increased during the ball phase. On a practical level, our findings indicate that pacas kept in rather barren standard farm environments may benefit from the inclusion of objects to increase investigation and general activity.

Agonistic behavior is very common in spotted pacas [21,24,28], although it is considered easy to manage in captivity [19,26]. In our study, we observed a reduction in agonistic behavior during the boomer ball phase compared with the control phases, as we predicted. Studies involving other species have also reported a decrease in aggressive behavior during object play behavior (e.g., *Pan paniscus* and *Pan troglodytes* [56]; *Tursiops truncates* [57]), while social play in adult animals can increase at times of social tension and high aggression risk (e.g., [58,59]). In our study, we provided one boomer ball per paca to allow access for all animals, thus minimizing potential resource disputes, although sometimes the pacas chased and tried to take the boomer ball from each other. In pigs (*Sus scrofa scrofa*), the ratio between the number of animals and objects used in environmental enrichment can be a limiting factor for the success of the procedure, as a low ratio can lead to competition and increased group aggression and frustration [60]. Thus, we infer that the low incidence of agonistic behavior during the ball phase suggests that the number of boomer balls in the pen was sufficient for all pacas and that the overall effect of their provision was positive for the animals.

Calls can indicate positive or negative emotions and provide information about animal welfare [47]. In our study, there were more bark emissions with a lower mean amplitude in the ball phase than the control phases, allowing us to infer that playing with boomer balls was associated with positive affective states. Lima et al. [30], when studying acoustic parameters as indicators of affective states and welfare in spotted pacas subjected to different affective state manipulations (negative, positive, ambiguous, and highly positive), found that barks with lower amplitudes were more prevalent in the assumed positive and highly positive valence treatments. However, they also found a decrease in the third quartile (Q75) frequency of bark calls in the positive situations [30], which was not evident in the current study. Given our other findings, this calls into question whether the specific energy distribution and fundamental frequency of bark calls provide reliable indications of positive emotional valence in spotted paca. The observation that introduction of boomer balls into the pens stimulated play behavior without increasing the mean amplitude of bark calls, as appears to happen in negative conditions indicating an elevated arousal state [30], suggests that this active play behavior was of a relaxed nature (cf. “having fun” [61]).

In general, play is more frequent in juveniles and becomes less frequent in adults [62], as in the case of the spotted paca [28]. Our study showed that this behavior can be induced, however, by enriching enclosures with objects such as boomer balls. This strategy can be used in farmed spotted pacas to reduce a state of boredom resulting from a predictable environment, as observed by other authors [63]. In adulthood, captive animals may particularly benefit from play stimulated by additional objects in their enclosures, when social play becomes less common [11,39,64].

One could argue that the period of 30 min a day with the boomer balls is relatively short. However, this practice was adopted as an alternative to leaving the balls permanently in the pens. This is because animals tend to lose interest in objects that are always present (e.g., [65]). Furthermore, the balls may be contaminated with animal feces, which could result in their rejection. Another potential criticism is that the animals used in the study had no prior experience with boomer balls. This could lead to animals rejecting these objects as a potential threat. However, as commented before, balls have been found to be harmless to animals and are highly valued for inducing play behavior in mammals [3]. Moreover, Nogueira et al. [29] reported that spotted pacas played with balls. In any case, probably due to their lack of previous experience with balls, the pacas tended to exhibit prolonged behavior of biting and touching these objects, as if testing them in some way. It is important to note that observed play behavior may not necessarily indicate positive welfare in spotted pacas. This behavior could be a rebound effect related to housing in a barren environment. Therefore, the occasional addition of balls for a brief period should not be solely relied upon as an indicator of their welfare. Additionally, the presence of play may only indicate a positive affective state during the time when the balls were present. Our study is limited to adult spotted paca; therefore, further studies are needed to verify whether juveniles exhibit more positive behavioral responses when playing with objects, as well as to determine if there are any differences in the acoustic parameters of their calls.

## 5. Conclusions

We conclude that the addition of boomer balls to the environment of farmed spotted pacas increased object play, and this was associated with increases in affiliative and exploratory behaviors, a reduction in agonistic interactions, and increases in bark emission at lower mean amplitudes, all of which, based on previous studies, can be considered markers of positive affective state. These indicators suggest that to achieve the potential welfare benefits of object play, this behavior must be stimulated in captivity because adult spotted pacas rarely show play behavior; provision of boomer balls may be one way to stimulate this.

## Figures and Tables

**Figure 1 animals-14-00078-f001:**
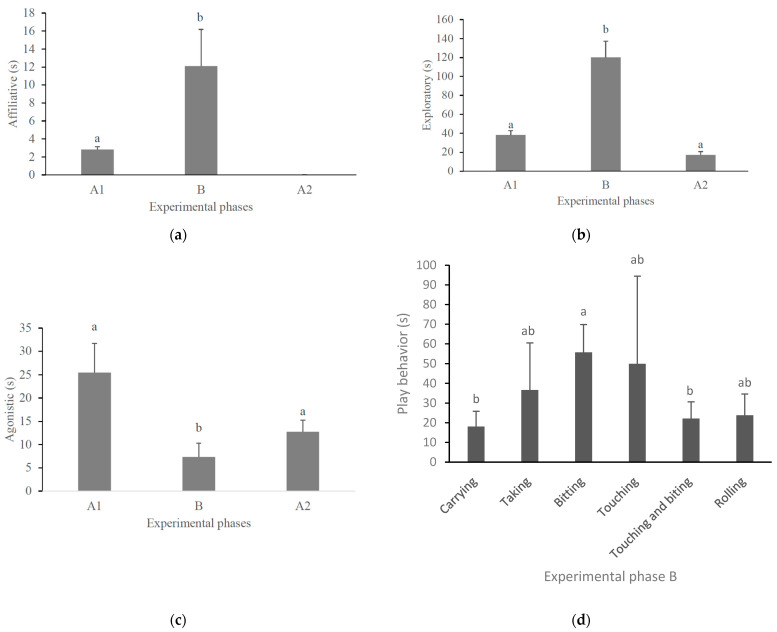
Mean time (+standard error) that pacas (N = 18) were observed in affiliative (**a**), exploratory (**b**), agonistic (**c**), and object play (**d**) behaviors during the experimental phases. Columns not sharing the same letter differed by Tukey’s post hoc test (*p* < 0.05).

**Figure 2 animals-14-00078-f002:**
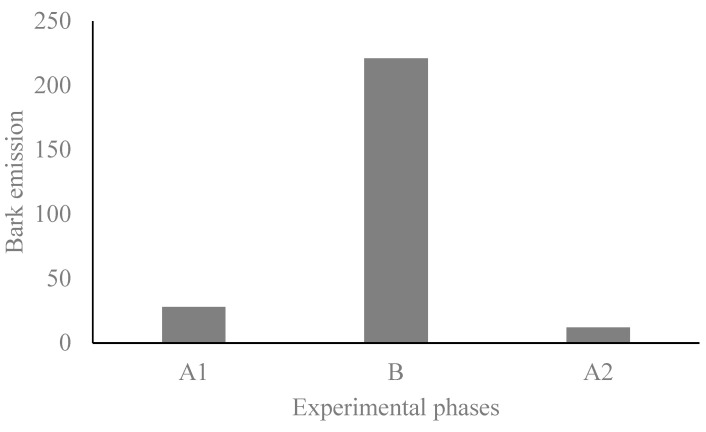
Total number of bark calls emitted according to the experimental phases.

**Figure 3 animals-14-00078-f003:**
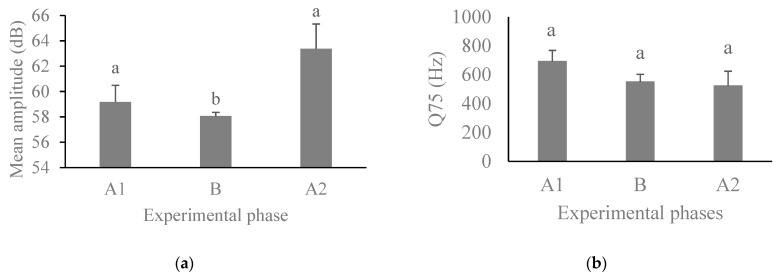
Mean amplitude (+standard error) (**a**) and mean frequency in the third quartile (Q75) (+standard error) (**b**) of barks emitted by pacas (N = 18) according to experimental phases. Columns with the same letter were not different by Tukey’s post hoc test (*p* < 0.05).

**Table 1 animals-14-00078-t001:** Behaviors observed during data collection.

Behavior *	Description	Reference
Affiliative	The individual touch with its snout the snout of another paca and/or may lie side by side.	[28]
Agonistic	The individual attacks another that does not respond to the aggression; the individual attacks another that responds aggressively, moving forward with fur raised and sometimes vocalizing.	[26,28]
Object play (with boomer ball) **	The individual picks up the ball and then moves around the pen with it (carrying); or it can take the ball to the shelter (taking); or it can bite the ball repeatedly (biting) and then touch the ball with one of its paws (touching) or it can do both movements (touching and biting); it can also roll the ball with its paw (rolling).	[38,39,40,41,42]
Exploratory	The individual sniffs the air with its head up. It may also sniff the ground or objects in the pen, except for the balls, with its head down.	[28]
Bark call	A call produced alone or in sequences of two to ten short elements (notes). ***	[24,30]

* The replacement of one behavioral state by the next was the criterion we used to determine the end of each behavioral state and the beginning of the next. ** We considered interactions with boomer ball as ‘play’ because they fulfil all Burghardt’s (2005) [2] criteria for play behavior (Appendix A: video clip that shows a paca playing with a boomer ball: biting, rolling, touching and biting). *** Details regarding how the observer determined that the focal individual was vocalizing are provided below.

**Table 2 animals-14-00078-t002:** Subtypes, mean duration (standard error) and occurrence of object play behaviors of spotted paca using the boomer ball (females N = 12 and males N = 6).

Play Behavior	Sex	Occurrence	Mean (s)
Carrying	Female	3	16.3 (9.1)
	Male	2	20.6 (18.5)
Taking	Female	1	60.5 (-)
	Male	1	12.7 (-)
Biting	Female	7	55.5 (21.2)
	Male	5	55.9 (19.2)
Touching	Female	1	5.5 (-)
	Male	5	94.5 (-)
Touching and biting	Female	2	25.3 (11.8)
	Male	6	14.2 (4.7)
Rolling	Female	2	23.8 (14.8)

**Table 3 animals-14-00078-t003:** Effect of sex, experimental phase, and the interaction of sex and experimental phase on the length of time pacas (N = 18) were observed engaging in affiliative, exploratory, and agonistic behaviors.

Behavior	Factor	F-Value	*p*-Value
Affiliative	Sex	F_1, 8.28_ = 2.16	0.179
	Phase	F_1, 7.83_ = 23.49	0.001
	Sex × Phase	F_1, 7.83_ = 4.94	0.058
Exploratory	Sex	F_1, 92_ = 0.76	0.386
	Phase	F_1, 92_ = 18.48	<0.001
	Sex × Phase	F_1, 92_ = 1.38	0.257
Agonistic	Sex	F_1, 24_ = 0.03	0.386
	Phase	F_2, 24_ = 3.93	0.033
	Sex × Phase	F_2, 24_ = 0.42	0.660

## Data Availability

The data presented in this study are available on request from the corresponding author. The data are not publicly available due to privacy restrictions.

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
