# Peer review of "Object Play as a Positive Emotional State Indicator for Farmed Spotted Paca (Cuniculus paca)"

_animals, 2023, doi:10.3390/ani14010078_

Round 1

Reviewer 1 Report

Comments and Suggestions for Authors

Dear authors, congratulations on your work. It is very interesting and makes an important contribution to the maintenance and well-being of pacas. The work is very well written and outlined. I have made some minor suggestions in the document, more to clarify certain parts. I believe that with these small changes, the work will be ready for publication. Congratulations again.

Author Response

Answers to the 1st reviewer’s comments:

Dear authors, congratulations on your work. It is very interesting and makes an important contribution to the maintenance and well-being of pacas. The work is very well written and outlined. I have made some minor suggestions in the document, more to clarify certain parts. I believe that with these small changes, the work will be ready for publication. Congratulations again.

Lima et al.: Thank you for your time and all your valuable comments. We followed your suggestion in the revised manuscript (ms) as detailed below.

L86. Delete

Lima et al.: We deleted the extra parenthesis. Thank you.

L156-158. Please, clarify. 10 min of observation/day for three days for all phases of the study. This would generate 30min of data. How 9h per phase were achieved? You repeated the three consecutive days of data collection in each phase until reach the 9h with intervals between these three days?

Lima et al.:  Thank you for your comment. There was a mistake in the first sentence. In the revised ms, trying to better explain our procedures, we reworded this, and the following sentences as follows:

L143-156: The behavior duration of each individual in the groups was observed for 10 minutes per day over three consecutive days for all phases of the study (described below). Thus, a total of 27 hours of data collection was conducted for all six groups, with 9 hours per phase. Further details are provided below.

L161: Maybe a space could be inserted here to separate table information from the following paragraph.

Lima et al.:  We added the requested space.

L214: What is the meaning of 35.5-s?

Lima et al.:  Thank you for your comment. In the revised ms, trying to better explain our results, we reworded this sentence as follows:

L214-216: Object play only occurred during the environmental ball phase (B), with a mean duration of 35.5 seconds (standard error (SE) = 6.4), regardless of the subtypes of object play.

Table 2: delete bold

Lima et al.:  Thank you, in the revised ms we deleted the bold.

Table 2: This means that no standard error was resulted from the analysis?

Lima et al.: When the standard error shows no value, it means that only one male or female exhibited the subtype of object play once. Thus, in the revised Table 2 we added the occurrences of each subtype of object play.

L225: Maybe a space could be inserted here to separate table information from the following paragraph.

Lima et al.:  We added the requested space.

L232: No comparions between the six play behaviors were conducted, correct? Is it possible to run this comparison?

Lima et al.: Thank you. In the revised ms we added the requested comparison in the Figure 1d, and change the text as follows:

L234-235: Among the subtypes of object play, they remained more time biting than carrying or touching and biting the boomer balls (Figure 1d).

L239: space between table and paragraph.

Lima et al.:  We added the requested space.

L239 Figure 2:  Delete the frame that appears around the picture.

Lima et al.:  We deleted the frame.

L247-248 Figure 3:  Delete the frame that appears around the picture.

Lima et al.:  We deleted both frames.

L331: delete? (in); suggestion: with a mean; suggestion: than in the

Lima et al.: Thank you. In the revised ms we accepted your corrections and reworded the sentences as follows:

L331-332: In our study there were more bark emissions with a mean lower amplitude in the ball phase than in the control phases, allowing us…

Lima et al.: Thank you again for your time and all your valuable comments.

Reviewer 2 Report

Comments and Suggestions for Authors

In general, this is an interesting article describing the novel topic of object play in adult pacas and relating it to affective states. The following are some suggestions for improving the manuscript.

Title, simple summary, abstract, last paragraph of the introduction, and conclusions: Please specify that the article is about object play, specifically.

L35. Please qualify. The observed play may have represented rebound behaviour related to housing in a barren environment. While likely indicating a positive state while playing, play would not be an indicator of good welfare in the absence of the balls.

L101. When explaining the Lima et al. (2022) findings, please specify that the results were from adult pacas of both sexes.

L117. Please change «and is classified as locomotor play» to «who reported on locomotor play». Otherwise, readers are led to expect that pacas only show locomotor play, and that this was the topic of the current study. It then comes as a surprise that the ethogram only includes object play.

L123. I think «observed» should be replaced by «proposed».

L147 and elsewhere. Remove h from the time. If included, place it after the time, following a space, not in the middle.

L156. The order … was (not were).

L159 and Table 1. Please clarify that it was behavioural durations that were recorded per 10 min observation.

Table 1. For each behaviour, add the criteria used to determine when each state ended and the next one started. Each behaviour should be described with focus on what the focal animal was doing. The individual … . Under agonistic behaviour, it is ambiguous what is meant by «that does not respond or reacts to». I think «or reacts» should be deleted. Under object play, please add a video clip illustrating a paca playing with a ball. Under exploratory, please clarify if objects included or excluded the balls. Under bark calls, please describe how the observer can determine when it is the focal pig who is calling. In the footnote, please change «it fulfils» with «they fulfil».

L169. I would replace «trying» with «aiming».

L172. Please add information about the pacas’ previous experience with balls.

L183 and elsewhere. You have already introduced experimental phases. It would be less confusing to readers if you stick with the «phases» terminology consistently rather switching between phase and treatment when referring to the same thing.

L200. I would replace «there was no way to compare» with «play could not be compared ...».

L202. Instead of «as well as their possible interactions», it should be «as well as their interaction».

Around L205. Please specify the method used to fit the models and assign the df.

Figure 1d. Please correct errors in the x axis labels: Touching ball with pall. Last label is cut-off.

Fig. 1d, Table 2, etc. When listing the different forms of play behaviour, use the same order consistently (ethogram, Fig 1d, Table 2, etc.). Also, once the behaviours have been defined in the ethogram, use precisely the same name for each behaviour throughout the rest of the text, instead of varying the names for the same behaviour (e.g. ethogram and Table 2: carrying the ball to the shelter; Fig. 1d: taking boomer ball). The bold text in Table 2 is then not needed.

L227. Compared to what?

L235. Given that the interaction was close to significance for affiliative behaviour, you might indicate how this behaviour tended to differ.

L254. suggesting (not indicating).

L256. Correction to “a positive”.

L269. This should be that.

L282-287. Consider sex differences in the play of species more closely related or cognitively comparable to pacas than the great ape examples given here.

L298. suggests (not indicates).

L335. subjected (not submitted).

End of discussion. Please add limitations of the study. 1. The animals were only observed for three 10 minute periods with balls. The potential consequences of leaving the balls permanently in the housing should be discussed. Animals typically lose interest in play objects when always present, and also when they get contaminated with faeces. 2. The impact of previous experience or lack of experience with the balls should be discussed, as this would likely affect the levels of play observed. 3. Adding balls briefly on occasion does not reliably indicate welfare of pacas all the rest of the time when they have no balls and are living in a barren environment. The presence of play likely indicates a positive affective state only during the time when the balls were present. 4. The results were limited to adults. Vocal measures associated with positive affect may be different in juveniles.

L361-362. Please qualify. The observed play may have represented rebound behaviour related to housing in a barren environment. While likely indicating a positive state while playing, play would not be an indicator of good welfare in the absence of the balls.

Comments on the Quality of English Language

Included in comments above.

Author Response

Answers to the 2nd reviewer’s comments:

In general, this is an interesting article describing the novel topic of object play in adult pacas and relating it to affective states. The following are some suggestions for improving the manuscript.

Lima et al.: Thank you for your time and all your valuable comments.

Title, simple summary, abstract, last paragraph of the introduction, and conclusions: Please specify that the article is about object play, specifically.

Lima et al.: Thank you. Throughout the text, we specify that our work is about object play.

L35. Please qualify. The observed play may have represented rebound behaviour related to housing in a barren environment. While likely indicating a positive state while playing, play would not be an indicator of good welfare in the absence of the balls.

Lima et al.: Thank you. Following your suggestion, we reworded the sentence as follows:

L31-34: Because the expression of object play was associated with a decrease in aggression, an increase in affiliative behavior, and an increase in low-amplitude barking, we suggest that object play can be used as a non-invasive indicator of positive emotional state in this species.

L101. When explaining the Lima et al. (2022) findings, please specify that the results were from adult pacas of both sexes.

Lima et al.: Thank you. Following your comment, we included the requested information as follows:

L98-100: In a recent study Lima et al. (2022) found that some vocalizations in adult pacas of both sexes are linked to their affective state and level of arousal. Specifically, the authors found that…

L117. Please change «and is classified as locomotor play» to «who reported on locomotor play». Otherwise, readers are led to expect that pacas only show locomotor play, and that this was the topic of the current study. It then comes as a surprise that the ethogram only includes object play.

Lima et al.: Thank you. Following your suggestion, we reworded the sentence as follows:

L104-106: In spotted paca, play has so far been reported only in captive young animals up to two months old (Sabatini and Paranhos da Costa, 2001), who reported on locomotor play, where the animals run alone in the enclosure and sometimes shake their heads…

L123. I think «observed» should be replaced by «proposed».

Lima et al.: Thank you. Following your suggestion, we reworded the sentence as follows:

L111-112: If object play behavior indicates positive affective states and welfare as proposed by Lawrence (1987), we would predict…

L147 and elsewhere. Remove h from the time. If included, place it after the time, following a space, not in the middle.

Lima et al.: We fixed it. Thank you.

L156. The order … was (not were).

Lima et al.: We fixed it. Thank you.

L159 and Table 1. Please clarify that it was behavioural durations that were recorded per 10 min observation.

Lima et al.:  Thank you for your comment. There was a mistake in the first sentence. In the revised ms, trying to better explain our procedures, we reworded this, and the following sentences as follows:

L143-146: The behavior duration of each individual in the groups was observed for 10 minutes per day over three consecutive days for all phases of the study (described below). Thus, a total of 27 hours of data collection was conducted for all six groups, with 9 hours per phase. Further details are provided below.

Table 1. For each behaviour, add the criteria used to determine when each state ended and the next one started. Each behaviour should be described with focus on what the focal animal was doing. The individual … . Under agonistic behaviour, it is ambiguous what is meant by «that does not respond or reacts to». I think «or reacts» should be deleted. Under object play, please add a video clip illustrating a paca playing with a ball. Under exploratory, please clarify if objects included or excluded the balls. Under bark calls, please describe how the observer can determine when it is the focal pig who is calling. In the footnote, please change «it fulfils» with «they fulfil».

Lima et al.:  Thank you for your comment. We followed your suggestions and completely reworked Table 1. We also included information regarding how the observer determined that the focal spotted paca was vocalizing as follows:

L179-182: Briefly, to another observer (AFL) was given a list of bark calls to analyze along with the corresponding timestamps in the video footage to conduct this analysis. The observer could easily identify the caller because of their natural characteristics, as explained above.

In the revised manuscript, we have included a supplementary video clip (S1) that shows a paca playing with a boomer ball, as requested.

L169. I would replace «trying» with «aiming».

Lima et al.: Following your suggestion we replace «trying» by «aiming» as follows:

L164-165: Aiming to avoid competition and encourage play behavior, each individual was given one ball.

L172. Please add information about the pacas’ previous experience with balls.

Lima et al.: Thank you, in the revised ms we explained that the animals had no previous experience with boomer balls as follows:

L165-166: The animals used in this study had no prior experience with balls.

L183 and elsewhere. You have already introduced experimental phases. It would be less confusing to readers if you stick with the «phases» terminology consistently rather switching between phase and treatment when referring to the same thing.

Lima et al.: Thank you, in the revised ms we replace treatment by experimental phases.

L200. I would replace «there was no way to compare» with «play could not be compared ...».

Lima et al.: Thank you, in the revised ms we reworded the sentence as follows:

L200: …play could not be compared between phases.  

L202. Instead of «as well as their possible interactions», it should be «as well as their interaction».

Lima et al.: In the revised ms we fixed it, thank you.

Around L205. Please specify the method used to fit the models and assign the df.

Lima et al.: In the revised ms we included the method used to fit the models as follows:

L196-199: We compared the time animals spent in each of the recorded behaviors (see Table 1) across experimental phases using mixed linear models (MLMs); one model for each behavior (affiliative, agonistic, and exploratory). The models were fitted to the data using the restricted maximum likelihood (REML) method.  

However, we were unable to assign degrees of freedom (df) for the analyses due to varying numbers of males and females performing each of the analyzed behavioral states. Therefore, the statistical software (Minitab 21.1) determined the df for each analysis.

Figure 1d. Please correct errors in the x axis labels: Touching ball with pall. Last label is cut-off.

Lima et al.: In the revised ms we deleted the term ball in all subtypes of object play to fix the x axis labels. Thank you.

Fig. 1d, Table 2, etc. When listing the different forms of play behaviour, use the same order consistently (ethogram, Fig 1d, Table 2, etc.). Also, once the behaviours have been defined in the ethogram, use precisely the same name for each behaviour throughout the rest of the text, instead of varying the names for the same behaviour (e.g. ethogram and Table 2: carrying the ball to the shelter; Fig. 1d: taking boomer ball). The bold text in Table 2 is then not needed.

Lima et al.: Thank you. In the revised ms we used both the same name and order of subtypes of object play. We deleted the bold text in Table 2 as well.

L227. Compared to what?

Lima et al.: We reworded the following sentences trying to better explain our results as follows:

L228-234: There was also variation in the expression of exploratory behavior and agonistic interactions depending on the experimental phase (Table 3). The post hoc tests showed that pacas displayed exploratory behavior for a longer duration during the ball phase (B) compared to the control phases (A1 and A2) (Figure 1b). On the other hand, pacas interacted agonistically for less time during the ball phase (B) than during the control phases (A1 and A2) (Figure 1c).

L235. Given that the interaction was close to significance for affiliative behaviour, you might indicate how this behaviour tended to differ.

Lima et al.: Thank you. In the revised ms we included the information that there was a tendency for males to show more affiliative behavior than females during the ball phase (B) as follows:

L235-239: Sex and the interaction between sex and experimental phase did not affect the time the pacas were observed in the behaviors analyzed (Table 3). However, during the ball phase (B), males showed a trend (F1, 7.83 = 4.94, P = 0.058, Table 3) to display affiliative behavior for a longer duration than females (males: mean = 20.8-s, SE = 9.0; females: mean = 6.9-s, SE = 2.0).

L254. suggesting (not indicating).

Lima et al.: Thank you. In the revised ms, we have replaced indicating with suggesting, as you suggested.

L256. Correction to “a positive”.

Lima et al.: In the revised ms, we fixed it. Thank you.

L269. This should be that.

Lima et al.: Thank you. In the revised ms we reinforced that this would be an expected result as follows:

L274-276: For example, in this species, as expected adding objects to the environment promotes increased possession of the objects by the dominant individual early in the animals’ exposure to the new objects, and later the objects are shared by the dominant individuals (Nogueira et al., 2011).

L282-287. Consider sex differences in the play of species more closely related or cognitively comparable to pacas than the great ape examples given here.

Lima et al.: Thank you. In the revised manuscript, we have rephrased these sentences to include reference to rodent species, as follows:

L284-288: Sex differences are well known for social play, specifically play fighting in rodents, but less investigated to date for object play. In rats and hamsters, some authors have reported that males display more social play than females, while others have reported no sex difference and others have suggested that these sex differences are related to experimental conditions (for a comprehensive review see Cooper et al., 2023).

L298. suggests (not indicates).

Lima et al.: Thank you. In the revised manuscript, we replaced indicates by suggests.

L335. subjected (not submitted).

Lima et al.: Thank you. In the revised manuscript, we replaced submitted by subjected.

End of discussion. Please add limitations of the study. 1. The animals were only observed for three 10 minute periods with balls. The potential consequences of leaving the balls permanently in the housing should be discussed. Animals typically lose interest in play objects when always present, and also when they get contaminated with faeces. 2. The impact of previous experience or lack of experience with the balls should be discussed, as this would likely affect the levels of play observed. 3. Adding balls briefly on occasion does not reliably indicate welfare of pacas all the rest of the time when they have no balls and are living in a barren environment. The presence of play likely indicates a positive affective state only during the time when the balls were present. 4. The results were limited to adults. Vocal measures associated with positive affect may be different in juveniles.

Lima et al.: Thank you very much. In the revised Discussion, we added limitations of our study. However, as explained before, there was a mistake when we explained the observation procedures. Each individual of the groups was observed for three 10 minutes periods with balls. In the revised ms, trying to better explain our procedures, we included the information that ‘(L162-163) Thus, in phase B we introduced three boomer balls to each group for 30 minutes daily’. In the revised Discussion, we included the reasoning that the period of 30 minutes a day with the boomer balls could be considered relatively short. We also included other points you highlighted as follows:

L353-372: One could argue that the period of 30 minutes a day with the boomer balls could be considered relatively short. However, this practice was adopted as an alternative to leaving the balls permanently in the pens. This is because animals tend to lose interest in objects that are always present (e.g. Kuczaj et al., 2002). Furthermore, the balls may be contaminated with animal feces, which could result in their rejection. Another potential criticism is that the animals used in the study had no prior experience with boomer balls. This could lead to animals rejecting these objects as a potential threat. However, as commented before, balls have been found to be harmless to animals and are highly valued for inducing play behavior in mammals (Fagen, 1981). Moreover, Nogueira et al. (2021) reported that spotted pacas played with balls. In any case, probably due to their lack of previous experience with balls, pacas tend to exhibit prolonged behavior of biting and touching these objects, as if testing them in some way. It is important to note that observed play behavior may not necessarily indicate positive welfare in spotted pacas. This behavior could be a rebound effect related to housing in a barren environment. Therefore, the occasional addition of balls for a brief period should not be solely relied upon as an indicator of their welfare. Additionally, the presence of play may only indicate a positive affective state during the time when the balls were present. Our study is limited to adult spotted paca, therefore, further studies are needed to verify whether juveniles exhibit more positive behavioral responses when playing with objects, as well as to determine if there are any differences in the acoustic parameters of their calls.

L361-362. Please qualify. The observed play may have represented rebound behaviour related to housing in a barren environment. While likely indicating a positive state while playing, play would not be an indicator of good welfare in the absence of the balls.

Lima et al.: Thank you. Based on your comments we reworded the Conclusion as follows:

L374-381: We conclude that the addition of boomer balls to the environment of farmed spotted pacas increased object play, and this was associated with increases in affiliative and exploratory behaviors, a reduction in agonistic interactions, and increases in bark emission at lower mean amplitudes all of which, based on previous studies, can be considered markers of positive affective state. These indicators suggest that to achieve the potential welfare benefits of object play, this behavior must be stimulated in captivity because adult spotted pacas rarely show play behavior; provision of boomer balls may be one way to stimulate this.

Lima et al.: Thank you again for your time and all your valuable comments.

Reviewer 3 Report

Comments and Suggestions for Authors

This is a straightforward and well written report on a less well studied species. It is nice to see a paper focused on an indicator of positive welfare, as this is neglected compared to investigations of negative welfare. It is nice that the findings could be replicated across six groups of pacas. However, it might have been beneficial to have at least one control group to show no effects during the same period of time.

The introduction provides sufficient background and rationale for the study.

Did you observe groups or individuals? You state that groups were observed (line 156) but you state that you used focal sampling (line 159). Can you clarify?

10 minutes per day per group seems insufficient. Can you justify this short sampling period?

Why do you think you observed no affiliative behavior in Phase A2?

Figures should show where the statistically significant differences occurred using asterisks.

The authors should note limitations in the discussion, which is otherwise well done.

Punctuation should be inside of quotations (e.g., line50, check throughout).

Author Response

Answers to the 3rd reviewer’s comments:

This is a straightforward and well written report on a less well studied species. It is nice to see a paper focused on an indicator of positive welfare, as this is neglected compared to investigations of negative welfare. It is nice that the findings could be replicated across six groups of pacas. However, it might have been beneficial to have at least one control group to show no effects during the same period of time.

The introduction provides sufficient background and rationale for the study.

Lima et al.:  Thank you for your time and valuable comments.

Did you observe groups or individuals? You state that groups were observed (line 156) but you state that you used focal sampling (line 159). Can you clarify?

Lima et al.:  Thank you very much for this comment. There was a mistake in the first sentence. In the revised ms, trying to better explain our procedures, we reworded this, and the following sentences as follows:

L143-146: The behavior duration of each individual in the groups was observed for 10 minutes per day over three consecutive days for all phases of the study (described below). Thus, a total of 27 hours of data collection was conducted for all six groups, with 9 hours per phase. Further details are provided below.

10 minutes per day per group seems insufficient. Can you justify this short sampling period?

Lima et al.:  As explained in the previous answer, there was a mistake in the first sentence. Therefore, we introduced the ball for 30 minutes in each day of the phase B. In the revised ms, trying to better explain our procedures, we included the information that ‘(L162-163) Thus, in phase B we introduced three boomer balls to each group for 30 minutes daily’. We limit enrichment with the ball to 30 minutes a day to avoid habituation as explained in the revised Discussion as follows:

L356-358: One could argue that the period of 30 minutes a day with the boomer balls could be considered relatively short. However, this practice was adopted as an alternative to leaving the balls permanently in the pens. This is because animals tend to lose interest in objects that are always present (e.g. Kuczaj et al., 2002). Furthermore, the balls may be contaminated with animal feces, which could result in their rejection. 

Why do you think you observed no affiliative behavior in Phase A2?

Lima et al.: Thank you for this comment. In the revised ms we try to explain this result as follows:

L300-303: Nevertheless, it seems reasonable that the absence of balls in phase A2, after three consecutive days of playing with this type of enrichment, may have negatively affected the animals’ mood. This may explain the lack of affiliative interactions during this phase.

Figures should show where the statistically significant differences occurred using asterisks.

Lima et al.:  Usually, asterisks are used to indicate significant differences. However, as we used Tukey's post hoc tests, we need to indicate which means differed. Therefore, we used letters to indicate the means that differed and those that did not. To explain that we added the following information above the graphs: Columns not sharing the same letter differed by Tukey’s post hoc test (P < 0.05).

The authors should note limitations in the discussion, which is otherwise well done.

Lima et al.:  Thank you. Following recommendations of the 2nd reviewer we added at the ended of the revised Discussion a paragraph on limitations of our study as follows:

L353-372: One could argue that the period of 30 minutes a day with the boomer balls could be considered relatively short. However, this practice was adopted as an alternative to leaving the balls permanently in the pens. This is because animals tend to lose interest in objects that are always present (e.g. [65]. Furthermore, the balls may be contaminated with animal feces, which could result in their rejection. Another potential criticism is that the animals used in the study had no prior experience with boomer balls. This could lead to animals rejecting these objects as a potential threat. However, as commented before, balls have been found to be harmless to animals and are highly valued for inducing play behavior in mammals [3]. Moreover, Nogueira et al. [29] reported that spotted pacas played with balls. In any case, probably due to their lack of previous experience with balls, pacas tend to exhibit prolonged behavior of biting and touching these objects, as if testing them in some way. It is important to note that observed play behavior may not necessarily indicate positive welfare in spotted pacas. This behavior could be a rebound effect related to housing in a barren environment. Therefore, the occasional addition of balls for a brief period should not be solely relied upon as an indicator of their welfare. Additionally, the presence of play may only indicate a positive affective state during the time when the balls were present. Our study is limited to adult spotted paca, therefore, further studies are needed to verify whether juveniles exhibit more positive behavioral responses when playing with objects, as well as to determine if there are any differences in the acoustic parameters of their calls.

Punctuation should be inside of quotations (e.g., line50, check throughout).

Lima et al.:  We fixed it. Thank you.

Lima et al.:  Thank you again for your time and valuable comments.